# Characterization of the Charge Heterogeneity of a Monoclonal Antibody That Binds to Both Cation Exchange and Anion Exchange Columns under the Same Binding Conditions

**DOI:** 10.3390/antib13030052

**Published:** 2024-06-30

**Authors:** Ming-Ching Hsieh, Jingming Zhang, Liangjie Tang, Cheng-Yen Huang, Yang Shen, Alice Matathia, Jun Qian, Babita Saxena Parekh

**Affiliations:** 1Analytical Sciences, Eli Lilly and the Company, Branchburg, NJ 08876, USA; 2Analytical Development, Eli Lilly and the Company, Indianapolis, IN 46221, USA; 3Antibody Technology, Eli Lilly and the Company, New York, NY 10016, USA; 4TS/MS Laboratories, Eli Lilly and the Company, Branchburg, NJ 08876, USA

**Keywords:** charge heterogeneity, anion exchange, cation exchange, monoclonal antibody, chemical labeling, hydrogen–deuterium (H/D) exchange mass spectrometry, molecular modeling

## Abstract

Therapeutic antibodies play an important role in the public healthcare system to treat patients with a variety of diseases. Protein characterization using an array of analytical tools provides in-depth information for drug quality, safety, efficacy, and the further understanding of the molecule. A therapeutic antibody candidate MAB1 exhibits unique binding properties to both cation and anion exchange columns at neutral pH. This uniqueness disrupts standard purification processes and necessitates adjustments in manufacturing. This study identifies that the charge heterogeneity of MAB1 is primarily due to the N-terminal cyclization of glutamine to pyroglutamine and, to a lesser extent, succinimide intermediate, deamidation, and C-terminal lysine. Using three approaches, i.e., deferential chemical labeling, H/D exchange, and molecular modeling, the binding to anion exchange resins is attributed to negatively charged patches on the antibody’s surface, involving specific carboxylic acid residues. The methodologies shown here can be extended to study protein binding orientation in column chromatography.

## 1. Introduction

Therapeutic proteins, particularly therapeutic monoclonal antibodies (mAbs), have played an important role in public healthcare and pharmaceutical markets. Currently, there are many antibody-related therapeutics, including mAbs, Fabs, pegylated Fabs, radio-labeled mAbs, antibody–drug conjugates or ADCs, bispecific mAbs, and IgG fusion proteins, that have been approved in the US/EU. Up to date in mid-2024, there are more than 130 therapeutic monoclonal antibodies approved in the US/EU and more than 20 are being reviewed by the EU or the FDA [1], not including 37 biosimilar therapeutic antibodies [2], and several monoclonal antibodies under EUA (emergency use authorization) for COVID-19 treatment [3]. Therapeutic antibodies offer several advantages, such as high specificity, less side effects, longer half-life, and wide applications. The monoclonal antibody market is expected to hit $425 billion by 2028 due to higher prevalence of cancer and the extensive R&D by research institutes and companies [4].

With the rapid emergence of therapeutic mAbs in the treatment of various diseases, ensuring the quality and efficacy is crucial in biopharmaceuticals to achieve the desired therapeutic effects. Monoclonal antibodies are typically produced in mammalian expression systems such as Chinese hamster ovaries (CHOs), human embryonic kidneys (HEKs) 293, or mouse myeloma cells (SP2/0, NS0) [5,6,7], and are subjected to many post-translational modifications (PTMs) during manufacturing processes, storage, and delivery [8,9]. The production of therapeutic monoclonal antibodies involves many advanced processes based on cell line types, bioreactor sizes, production titers, and purification efficiency. To streamline development efforts, a platform approach is suggested [10]. For instance, after the initial capture step with Protein A chromatography, ion exchange chromatography is usually employed to further purify mAbs by removing process- and product-related impurities [11]. Anion exchange chromatography (AEX) is commonly used to eliminate host cell DNA, adventitious viruses [12], host cell proteins, residual Protein A, and proteins in culture medium such as BSA. Both anion and cation exchange chromatography (CEX) can potentially remove aggregates [13].

In addition to its use in manufacturing processes, ion exchange chromatography is widely used for the analytical characterization of charge variants of monoclonal antibodies [14], separating these biomolecules according to their net surface charges [15] as well as their conformation [16]. Charge variants can arise from post-translational modifications (PTMs) during biosynthesis, such as charged oligosaccharides (sialic acids) in the carbohydrate moiety, incomplete signal peptide cleavage, and C-terminal lysine clipping. PTMs can also result from non-enzymatic reactions during storage, transportation, and temperature excursion, such as N-terminal pyroglutamation, deamidation, isomerization, oxidation, acid-base hydrolysis, and glycation (see reviews in [8,9,14,17,18]). Charge variants are typically categorized into three main groups, i.e., acidic, main, and basic charged groups, based on the elution profile instead of their apparent isoelectric point (pI) [19]. Acidic variants, which eluted earlier in a cation exchange chromatography, are more acidic than the main peak. Basic variants are eluted later due to more interactions between the positive ions and the resin. In general, the PTMs that increase “negative” ions or less “positive” ions, such as sialic acid, deamidation, glycation, N-terminal pyroglutamation from glutamine, and arginine modification by methylglyoxal [20] are attributed to the acidic variants. Conversely, PTMs that increase “positive” ions or less “negative” ions, such as C-terminal lysine, C-terminal amidation, isomerization of aspartate, and succinimide intermediate formation, contribute to the basic variants [21,22]. Sometimes other PTMs, such as oxidation, disulfide reduction, cysteinylaton, and N-terminal pyroglutamation from glutamic acid, that do not introduce charge to the molecule, are enriched in either acidic or basic variants depending on the conformational change of the protein that alters the binding to the CEX column [18,23,24]. 

The separation mechanism of ion exchange chromatography (IEC) is based on the reversible adsorption of charged protein molecules to the immobilized exchange groups of opposite charge at a specific pH, where the protein is charged according to its isoelectric point (pI). Given the neutral to basic pI values of most monoclonal antibodies, it is common practice in the industry to analyze their charged variants using cation exchange (CEX) chromatography with either salt or a pH gradient. Although antibodies can be purified using either anion or cation exchange chromatography [15], the initial binding pH values are typically different to ensure the positive or negative net charge for cation exchanger or anion exchanger binding, respectively. Binding to both cation and anion exchangers under identical pH conditions is a rare characteristic for a protein. There is limited information in this regard; an instance involving an RNA polymerase holoenzyme has been reported to bind to both a cation exchanger (Bio-Rex 70) and anion exchanger (DEAD Sephadex A25) in 0.1 M KCl buffer [25]. The RNA polymerase holoenzyme consisted of many subunits (β, β′, σ, α) and the entire holoenzyme is big with a molecular weight over 450 kDa. It is possible that the binding regions to cation and anion resins may be on different subunits. 

In the initial stages of process development for MAB1, an IgG_1_ therapeutic antibody, the standard purification scheme, i.e., Protein A purification, anion exchange flow through and further cation exchange purification, did not work. MAB1 unexpectedly adhered to the anion exchanger at the neutral pH despite high pI values of MAB1. Our investigation into MAB1’s charge heterogeneity, using an array of analytical techniques, revealed that the major difference for the charge variants was the N-terminal pyroglutamate. More interestingly, MAB1, out of more than a dozen mAbs developed in-house, binds both cation and anion exchangers under identical binding conditions (10 mM sodium phosphate pH 7.2 ± 0.1). Such a characteristic is likely attributable to a distinct local area that remains negatively charged even below its apparent pI values, posing significant challenges to the downstream purification process.

The putative binding site(s) of MAB1 to anion exchange resin are likely the carboxylic side chains of glutamic acid (Glu) and aspartic acid (Asp) residues. Analysis of the primary sequence reveals that MAB1 has a slightly higher percentage of Glu and Asp residues compared to other mAbs, particularly in the variable regions. To identify these binding sites under pH conditions below its apparent pI, three methods were utilized: differential chemical labeling, H/D exchange mass spectrometry, and molecular modeling. Chemical labeling has previously been used to study the binding orientation of lysozyme on cation exchange resin [26], to identify a positive charge patch of a mAb [27], and to characterize the structure of another monoclonal antibody by footprinting [28]. H/D exchange mass spectrometry has been successfully applied to examine antibody conformations in its native state [29] or after modifications [30]. Additionally, molecular modeling serves as a powerful tool to study antibody structures [31]. Here we demonstrated the feasibility to use these three methods (differential chemical labeling, H/D exchange mass spectrometry, and molecular modeling) to understand the binding interactions between MAB1 and the anion exchanger.

## 2. Experimental

### 2.1. Chemicals and Reagents

The fully human monoclonal antibody, MAB1, for this study was produced in Chinese hamster ovary (CHO) cells and purified by Process Development Group at Lilly Branchburg, NJ, USA. Trypsin (sequencing grade TPCK-modified) was obtained from Promega Corporation (Madison, MI, USA). Dithiothreitol (DTT), iodoacetamide, glycine ethyl ester (GEE), D_2_O, deuterated phosphoric acid, Tris(2-carboxyethyl)phosphine hydrochloride (TCEP•HCl), and formic acid were obtained from Sigma-Aldrich (St. Louis, MO, USA). 1-ethyl-3-(3-dimethylaminopropyl)carbodiimide (EDC) and N-hydroxysuccinimide (NHS) were obtained from Pierce Biotechnology (Rockford, IL, USA). The NAP-5 cartridge and Q Sepharose Fast Flow resin were obtained from Cytiva (Marlborough, MA, USA). Carboxypeptidase B (CPB) was obtained from Roche Diagnostics (Indianapolis, IN, USA). Guanidine hydrochloride (GudHCl) and acetonitrile were obtained from J. T. Baker (Avantor Inc., Philadelphia, PA, USA). Sodium phosphate, acetic acid, sodium chloride, and Tris-HCl pH 7.5 buffer were purchased from J. T. Baker, Millipore Sigma (St. Louis, MO, USA), and Fisher Scientific (Waltham, MA USA), respectively. 2-aminobenzoic acid (2-AA), 2-picoline borane complex, trifluoroacetic acid (TFA), and ammonium formate were purchased from Millipore Sigma (St. Louis, MO, USA). 

### 2.2. Ion Exchange Chromatography

#### 2.2.1. Cation Exchange Chromatography (CEX)

Protein samples were desalted using a Sephadex G-25 cartridge (NAP-5), following the manufacturer instructions prior to column chromatography. Cation exchange chromatography was conducted with an analytical weak cation exchange column (ProPac WCX-10, 250 × 4.0 mm, 10 µm, Thermo Fisher Scientific (Waltham, MA USA) using an Agilent 1100 series HPLC (Agilent Technologies, Wilmington, DE, USA). Protein (total of 50 μg) was monitored at 214 and 280 nm. Buffer A was 10 mM sodium phosphate, pH 7.2 ± 0.1, and Buffer B was 200 mM NaCl in 10 mM sodium phosphate buffer, pH 7.2 ± 0.1. The gradient started from 100%A to 15%B in 60 min at a flow rate of 0.5 mL/min. The column temperature was kept at 40 °C and was equilibrated with Buffer A for 15 min prior to next injection.

#### 2.2.2. Anion Exchange Chromatography (AEX)

Protein samples were also desalted using the NAP-5 cartridge prior to anion exchange chromatography (ProPac WAX-10, 4 × 250 mm with a guard column, ProPac WAX-10G, 4 × 50 mm, 10 µ, Thermo Fisher Scientific). Buffer A was 10 mM sodium phosphate, pH 7.2 ± 0.1, and Buffer B was 1 M NaCl in 10 mM sodium phosphate buffer, pH 7.2 ± 0.1. The gradient started from 10%B to 40%B in 90 min with a flow rate of 0.5 mL/min. The column temperature was kept at 30 °C and was equilibrated with Buffer A for 15 min prior to next injection.

### 2.3. IEF Gel Electrophoresis

Isoelectric focusing gel electrophoresis was performed using the Multiphor™ Electrophoresis system (GE Healthcare, Piscataway, NJ, USA). The samples were desalted using the NAP-5 cartridge. Total protein of 10 μg from the desalted samples was applied on the IEF agarose pH 6.5–10 gels (Lonza, Wakersville, MD, USA). It was allowed to focus for 110 min at 1000 V and 25 W. The IEF gel was then fixed in 10% trichloroacetic acid for 10 min and stained with Coomassie staining solution after washing the gel with deionized water to remove ampholytes.

### 2.4. Imaged Capillary Isoelectric Focusing (cIEF)

Imaged cIEF was performed using an iCE280 system with Alcott autosampler (Protein Simple, San Jose, CA, USA). All cartridges and solutions, including methyl cellulose, electrolytes, and pI makers, were purchased from Protein Simple. The desalted antibody sample (1 mg/mL, 60 µL) was mixed with 40 µL mixture comprising 0.875% methyl cellulose, 10% Pharmalyte^®^ 3–10 (Cytiva, Marlborough, MA, USA), pI makers of 5.85 and 9.5, and injected through the autosampler. The protein was focused for 1 min at 1500 V and for 10 min at 3000 V. After focusing, an electropherogram was taken with the CCD camera at 280 nm. 

### 2.5. Carboxypeptidase B (CPB) Treatment 

Carboxypeptidase B was reconstituted with Millipore water to make a 5 mg/mL stock. The enzyme stock was aliquoted and stored at –20 °C in a freezer upon use. The enzyme to substate ratio is 1:25 (w/w) and incubation was allowed at 37 °C for 3 h. The treated sample was directly subjected for CEX analysis. 

### 2.6. N-Linked Oligosaccharide Profiling with HILIC Analysis

N-linked oligosaccharides of MAB1 were released by PNGase F (Agilent Technologies, Santa Clara, CA, USA) in the reaction buffer for 18 h at 37 °C. The released N-linked oligosaccharides were derivatized with 2-AA with 2-picoline borane complex at 70 °C for 90 min. The derivatized glycans were purified by a solid phase extraction (SPE) cartridge (Supelco Discovery DPA-65, 50 mg, 1 mL). The N-linked oligosaccharides were eluted by 0.5 mL of 20% acetonitrile and stored at −20 °C in a freezer.

The purified 2-AA-labeled glycans were analyzed by UHPLC (1290 series, Agilent Technologies, Stata Clara, CA, USA) using an Acquity BEH Glycan column (1.7 µm 2.1 × 150 mm, Waters Corporation, Milford, MA, USA) with 100 mM ammonium formate as mobile phase A and acetonitrile as mobile phase B. The gradient started with 22.5%B to 45%B in 40 min. The column temperature was kept at 60 °C with the flow rate of 0.5 mL/min.

### 2.7. Tryptic Peptide Mapping

Monoclonal antibody MAB1 samples (~20 µg) were brought to complete dryness under vacuum. The proteins were denatured and reduced by guanidine hydrochloride and DTT in 50 mM Tris–HCl at 50 °C for 30 min. After denaturation and reduction, the proteins were alkylated with iodoacetamide for 30 min in darkness. The reaction mixture was then diluted with 50 mM Tris–HCl pH 7.5. The trypsin digestion was performed at 37 °C for 3 h using an enzyme:protein ratio of 1:10 (w/w). The reaction was terminated by adding 1 µL 20% TFA. The resultant peptides were resolved in a C18 reverse-phase column (Aeris widepore XB-C18, 200 Å, 3.6 µ, 250 × 2.1 mm, Phenomenex, Torrance, CA, USA) using an Agilent HPLC 1260 series (Agilent Technologies, Santa Clara, CA, USA) interfaced with an Exploris 480 mass spectrometer (Thermo Fisher Scientific, San Jose, CA, USA) equipped with an electrospray source. The temperature of the ion transfer tube was set at 275 °C. The spray voltage was set at 3.9 kV. The digested sample was eluted with a gradient from 0% solvent B to 40% solvent B (solvent A is 100% H_2_O with 0.05% TFA, solvent B is 100% acetonitrile with 0.04% TFA) for 50 min at a flow rate of 0.2 mL/min.

For analysis of chemical labeling with EDC, MAB1 samples were dried under vacuum. After denaturation and reduction, the proteins were alkylated and dialyzed against 20 mM Tris pH 7.0 at 4 °C for 2 h. The dialyzed solution was collected, and trypsin digestion was performed at 37 °C for 3 h using an enzyme:protein ratio of 1:20 (w/w). The reaction was terminated by adding 5 µL 50% TFA. The resultant peptides were resolved in a C18 reverse-phase column (Zorbax C18 300SB, 300 Å, 5 µ, 150 × 4.6 mm) using an Agilent HPLC 1200 series (Agilent, Wilmington, DE) interfaced with an LTQ ion trap mass spectrometer (Thermo Fisher Scientific, San Jose, CA, USA) with capillary temperature at 300 °C, sheath gas flow at 25 units, auxiliary gas flow at 5 units, and the source voltage at 4.6 kV. The digested sample was eluted with a gradient from 98% solvent A (0.1% TFA) to 40% solvent B (100% acetonitrile, 0.085% TFA) for 90 min at a flow rate of 0.5 mL/min. 

### 2.8. Top-Down Mass Analysis

Monoclonal antibody MAB1 (~100 µg) was diluted to 2 mg/mL. The proteins were deglycosylated using PNGase F overnight at 37 °C. The proteins (56 µg) were reduced with 50 mM DTT for 60 min at 50 °C with the presence of 2 M guanidine hydrochloride. The deglycosylated and reduced protein samples were resolved in a size exclusion column (TSKgel SW_XL_ 300 × 7.8 mm, Tosoh Bioscience, King of Prussia, PA, USA) using an Acquity UPLC H Class (Waters, Milford, MA, USA) interfaced with a SYNAPT G2-Si mass spectrometer (Waters, Milford, MA, USA). The HPLC was run for 50 min at a flow rate of 0.3 mL/min (20% acetonitrile in water with 0.1% TFA). The mass spectrometer capillary voltage was set at 1.5–2.5 kV with sampling cone at 150 and source offset at 100. The source and desolvation temperature were set at 150 °C and 400 °C, respectively. The cone gas was 50 L/h and desolvation gas was 600 L/h. The nebulizer was set at 6.5 bar. The intact MS was run with sensitivity mode under positive polarity. The m/z range of detection was set at 500–5000, and the scan rate was 1 spectra/second. The molecular weight for the peaks was deconvoluted by the MaxEnt1 function provided by the MassLynx software, Version 4.1 (Waters).

### 2.9. Chemical Labeling

Labeling of antibody carboxylate amino acid (D and E) side chains or C-termini with primary amines (glycine ethyl ester, GEE) was achieved by activating the carboxylate group with carbodiimide (EDC, 1-ethyl-3-(3-dimethylaminopropyl)carbodiimide) and stabilizing the unstable amine-reactive product with NHS (N-hydroxysuccinimide) to form semi-stable amine-reactive NHS ester. Briefly, EDC and NHS (ratio of 1:2.5) were added to the antibody sample in 10 mM sodium phosphate buffer, pH 7.0, with various EDC to antibody carboxylate ratios (1:62, 1:1, and 10:1) at room temperature with constant mixing. After 15 min, GEE (GEE to EDC ratio of 1:1) was added to the reaction at room temperature with constant mixing for 2 h. Then, all labeling chemicals (EDC, NHS, and GEE) were removed by buffer exchange using Amicon^®^ Ultra-4 centrifugal filter units (Millipore, Billerica, MA, USA). The general procedural scheme is illustrated in Figure 1. 

### 2.10. Hydrogen–Deuterium Exchange Mass Spectrometry (H/D EX MS) Analysis

The experimental design at ambient temperature (21 ± 0.5 °C) is illustrated in Figure 2 and briefly described below.

An MAB1 stock solution (13.4 mg/mL) was buffer-exchanged to sodium phosphate buffer (8 mM, pH 7.0 in 100% H_2_O) using a Microcon centrifugal filter with 30,000 MWCO (Ultracel YM-30, Millipore Sigma (St. Louis, MO, USA). The control sample of MAB1 was prepared by adding 7.5 µL of MAB1 solution containing phosphate buffer (in 100% H_2_O) to a 150 µL of Q Sepharose fast flow (FF) resin slurry (75% slurry in phosphate buffer) in an empty cartridge (Pierce). The cartridge containing the Q Sepharose FF resin and MAB1 was then centrifuged (~1000× *g*) for 5 s to remove all liquid. The same centrifugation force and duration were implemented for all subsequent procedures involving centrifugation. The cartridge was subsequently washed with 100 µL phosphate buffer twice. After washing, the phosphate buffer containing 80% D_2_O was added to the Q Sepharose FF resin/MAB1 to initiate H/D exchange. At the specific time points (i.e., 150, 500, 1500, and 5000 s), deuterium containing buffer was removed by centrifugation. The cartridge was then washed twice with 100 µL of phosphate buffer in H_2_O and centrifuged to remove liquid. An aliquot of 55 µL of an ice-chilled quench buffer (5.8 M GuHCl/0.8 M TCEP solution in H_2_O, pH 2.5) was added to quench the reaction by bringing down pH and temperature, and to elute the protein from the resin. The eluate was collected in a clean container after centrifugation and immediately frozen on dry ice and kept in a −70 °C freezer before data acquisition.

On the other hand, the exchanged sample (Ex-sample) was prepared by mixing 7.5 µL of MAB1 with 30 µL phosphate buffer containing 100% D_2_O. After the exchange for a specific time (150, 500, 1500, and 5000 s), the exchanged sample was loaded to 150 µL Q Sepharose FF resin equilibrated with phosphate buffer containing 80% D_2_O. The sample was treated the same way as described above for the preparation of the control samples, including steps to remove D_2_O buffer, wash by phosphate buffer with 100% H_2_O, quench the exchange reaction, collect the eluates, and freeze on dry ice. 

The maximum deuterated sample was prepared by mixing 4 µL of a 13.4 mg/mL MAB1 and 16 µL of a 100 mM TCEP solution in D_2_O. Such a mixture was incubated for 20 h at ambient temperature. The fully deuterated sample was then mixed with 30 µL of an ice-chilled buffer (5.8 M GuHCl/0.8 M TCEP solution in H_2_O, pH 2.5) to quench the H/D exchange. The quenched sample was then frozen on dry ice prior to test. The frozen sample was defrosted at ambient temperature and immediately injected onto a pepsin column (packed in-house, with ~100 µL bed volume). The on-column digestion was performed for 10 min on ice. After digestion, the sample was pumped onto a trap column (packed with POROS 10 R2 resin, Thermo Fisher Scientific, Waltham, MA, USA) and desalted using 0.1% formic acid at a flow rate of 300 µL/min for 30 s. The peptides were separated using a C18 column (Jupiter 1.0 × 50 mm, 5 µm, Phenomenex) with a linear gradient of 10% to 40% buffer B over 28 min (buffer A, 0.1% formic acid in water; buffer B, 95% acetonitrile in 0.1% formic acid) at a flow rate of 100 µL/min. All columns and connected capillaries were immerged under ice to minimize back-exchange during on-column digestion, desalting, and reversed-phase (RP) separation. Mass spectrometric data were acquired with a Thermo Finnigan LCQ Deca XP mass spectrometer (Thermo Fisher) using the capillary temperature set at 225 °C.

### 2.11. H/D Exchange Data Analysis

Prior to H/D exchange experiments of MAB1, the pepsin digestion conditions were optimized using non-deuterated MAB1 to achieve better sequence coverage of the protein. Data-dependent acquisition MS/MS and SEQUEST were used for the identification of digested fragments. DXMS software (Sierra Analytics, Inc., Modesto, CA, USA) was used to calculate the centroid mass value of each identified peptide with or without deuteration. Corrections for back-exchange were made using the equations below [32]:$$\mathrm{Deuteration}\;\mathrm{level}\;(\%)=\frac{m(P)-m(N)}{m(M)-m(N)}\times 100$$
$$\mathrm{Deuterium}\;\mathrm{recovery}\;(\%)=\frac{m(M)-m(N)}{\mathrm{MaxD}}$$
where m(P), m(N), and m(M) are the centroid value of partially deuterated, nondeuterated, and maximum-deuterated peptides, respectively. MaxD is the maximum deuterium level calculated by subtracting two and the number of prolines in the third or later position of the peptide from the number of amino acids in the peptide of interest. This is based on the assumption that the first two amino acids of the derived peptides cannot retain deuterium after RPLC separation. [33]. The deuterium recovery of a maximum-deuterated sample in this study was approximately 60% on average. The reported sequence number of each peptide corresponds to the actual sequence of the peptide without truncation of the first two amino acids that are expected to carry no deuterium.

### 2.12. In Vitro Binding Assay

In the binding assay of MAB1 on Biacore C (Cytiva, Marlborough, MA, USA), the innate antigen, prepared in-house, was immobilized onto a CM5 Sensor Chip using amine coupling. A calibration curve was generated by injecting the MAB1 product standard that spans from 78–2000 ng/mL onto the immobilized surface. The samples were tested at five different concentrations from 158 ng/mL to 800 ng/mL based on the nominal protein concentration determined previously. The concentration of MAB1 in a sample was calculated from its Resonance Units (RU) by comparison to the RU of solutions containing known amounts of MAB1 standard, using 4-parameter regression analysis. The potency values, reported as % binding activity, were calculated from the mean calculated concentration of MAB1 from the five dilutions (determined from comparison to the calibration curve) divided by the protein concentration (measured by A280) × 100.

### 2.13. Molecular Modeling

The structural model of the MAB1 Fv region was generated by a standard homology modeling procedure. VL and VH domains of MAB1 sequences were used to search against the BLAST database to locate the corresponding structure templates with the highest sequence identities. The VL of PDB 2R56 and the VH of PDB 3D69 were selected as structure templates for the VL (with 90% sequence identity) and the VH domains (with 83% sequence identity) of MAB1, respectively. The SWISS-MODEL server [34] was used to generate individual structural models for these domains from templates. The proper assembly of the modeled VL and VH domains into Fv was achieved by superposing the coordinate of the MAB1 VH model to the VH domain structure of PDB 2R56 by the program DaliLite [35]. The RMSD (root mean square deviation) of 118 Cα atoms structurally aligned is 0.6 Å. The Fv model for MAB1 was then examined in the program Coot [36]. Some residue sidechains with close proximities at the VL–VH interface were manually adjusted and locally refined. The program Pymol [37] was used to generate the structure rendering. The electrostatic surface potential of the model was calculated through the program APBS [38] as a plug-in in Pymol.

## 3. Results and Discussion

### 3.1. Characterization of MAB1 Charge Heterogeneity

MAB1 is an IgG_1_ monoclonal antibody produced in the CHO cell line. It has several charge isoforms characterized by IEF gel electrophoresis and by imaged capillary IEF in the pI range of 7.5 to 8.6 (Figure 3). Due to the pI of 7.5 to 8.6, monoclonal antibody MAB1 charge heterogeneity was evaluated in CEX chromatography in the phosphate buffer systems at pH 7.2 to ensure the overall positive charge for binding. It was fractionated into eight fractions by a shallow gradient with a main peak flanking with two acidic peaks and five basic peaks (Figure 4A blue trace). The eight fractions were collected, concentrated, and reinjected to check the purity, as shown in Figure 4B.

One of the most common charge variants can be attributed to the C-terminal lysine for monoclonal antibodies. The C-terminal lysine is not engineered out from the cDNA of MAB1, and therefore it contributes to the basic variants. The treatment of carboxypeptidase B (CPB) revealed that fractions B3 and B5 contained C-terminal lysine (Figure 4 red trace). CPB seemed to remove all C-terminal lysine in B5, but some other charge variant(s) remained in B3. C-terminal lysine was also observed from peptide mapping coupled with mass spectrometry analysis. Both B3 and B5 contained approximately 40% of C-terminal lysine (Table 1). The remaining components in B3 and B5 may be due to the succinimide intermediate observed, as described in the next section.

Peptide mapping coupled with mass spectrometry is a widely used technique for identifying and semi-quantifying peptides and their possible modifications. The tryptic peptides can be separated by reversed-phase HPLC and the identity of the peptides can be subsequently confirmed by mass spectrometry. The MAB1 peptide mapping profile showed that the pyroglutamate at the N-terminus of the heavy chain was a dominant modification for MAB1 but is less prominent in basic peaks B2–B5 (Table 1), especially for B2 and B4. It is expected that when glutamine is converted to pyroglutamate, it becomes more acidic due to the loss of the N-terminal primary amine [9]. In both the acidic and the main peaks, N-terminal pyroglutamate was nearly 100%, while deamidation at the constant region of the heavy chain was predominantly observed in the acidic peaks. Fraction B1 exhibited a relatively higher amount of succinimide intermediate in aspartate residues in the heavy chain variable region (Table 1). This succinimide intermediate is unstable and readily hydrolyzes to become aspartate or isoaspartate during sample preparation. The presence of succinimide intermediate in B1 and B3 was further confirmed when the MAB1 sample was buffer-exchanged to alkaline conditions (pH 8 and pH 9) and incubated at 37 °C overnight (approximately 18 h). This treatment resulted in the conversion of succinimide to aspartate or isoaspartate, while it remained stable at slightly acidic and neutral pH (Figure 5). The disappearance of B1 and B3, along with the increase in the A1 peak, suggested the presence of succinimide intermediates, which are known to be short-lived in alkaline conditions, as documented in the literature [39].

Another potential source of charge heterogeneity is from the sialic acid in the carbohydrate moiety of the antibody. The N-linked glycans were hydrolyzed by PNGase F and labeled with 2-aminobenzoic acid (2AA) and separated by hydrophilic interaction chromatography (HILIC). The analysis indicated an absence of sialic acid in the MAB1 carbohydrate moiety, as depicted in Figure 6. Consequently, sialic acid can be ruled out as a factor influencing the charge heterogeneity. 

The top-down mass analysis revealed that the molecular mass of deglycosylated MAB1 was approximately 145,067.5 Daltons (theoretical mass of 145,058 Daltons) for the *des*-K variant of heavy chains. One C-terminal lysine was observed at 145,195 Daltons (Figure 7A); however, two C-terminal lysine charge variants were not observed in MAB1, consistent with the peptide mapping data (Table 1). The molecular mass of the glycosylated heavy chain was approximately 50,719.5 Daltons (theoretical mass of 50,720.7 Daltons), indicating the major glycoform of MAB1 is G0F (Figure 7B), consistent with N-linked oligosaccharide profiling results (Figure 6). Another glycoform with approximately 50,881 Daltons is the G1F glycoform (theoretical mass of 50,882.8 Daltons). The top-down mass spectrometry analysis also proved the absence of sialic acid in MAB1, therefore indicating no contribution to charge variants.

Based on the aforementioned analytical results, the charge heterogeneity of MAB1 mainly consists of N-terminal pyroglutamate, succinimide intermediate, deamidation, and C-terminal lysine, with no presence of sialic acid. The charge heterogeneity of MAB1 was not atypical in therapeutic antibodies. Further investigation was needed to understand why MAB1 binds to both cation and anion exchangers in the same neutral conditions.

### 3.2. Monoclonal Antibody MAB1 Binds to Anion Exchange and Cation Exchange Resins

Monoclonal antibody MAB1 was found to bind to the anion exchange resin during the development of the purification process. Anion exchange (Q Sepharose Fast Flow) chromatography in antibody purification is used to remove manufacturing-related impurities such as host cell proteins, DNAs, viruses, and endotoxins in a flow-through mode [11]. To ensure the flow-through mode, the pH of the protein solution is set below the pI value of the antibody, resulting in a net positive charge on the protein (e.g., ~pH 7.2), while impurities are retained. The binding ability of MAB1 to Q Sepharose resin interrupted the purification scheme in the manufacturing processes. The binding to ion exchange resin is mainly through electrostatic interactions [40], suggesting that MAB1 may have a local negative-charged area that interacts with anion exchange resin at neutral pH. 

MAB1 was separated into eight peaks by CEX (Figure 4) at pH 7.2. The antibody was also separated by an AEX column, with fewer peaks observed, using the same phosphate buffer systems at pH 7.2 (Figure 8A). The isolated CEX fractions were analyzed by AEX in the identical pH buffer systems. Interestingly, the AEX peaks did not align with the CEX elution profile, as shown in Figure 8B, suggesting different interactions between the protein and ion exchangers.

MAB1 binds to both strong anion exchangers (AEXs) like Q Sepharose Fast Flow and weak AEXs like ProPac WAX-10. The initial chromatographic conditions of ProPac WAX-10 are identical to those for ProPac WCX-10, using a 10 mM sodium phosphate buffer at pH 7.2 ± 0.1. However, the elution for AEXs required a much higher salt concentration (see Experimental Section 2.2.2). Despite the different functional groups of these two anion exchange resins, Q Sepharose resin was chosen to study the binding site(s) of MAB1 due to the feasibility and use in the purification process. Q Sepharose Fast Flow features quaternary amine ligands attached to macroporous polymer (agarose)-derived spherical beads, while the ProPac WAX-10 column has a tertiary amine functional group grafted on nonporous core particles. Both resins are hydrophilic, minimizing undesirable hydrophobic interactions with proteins. The ProPac WAX-10 column is packed with uniform 10 µm particle size beads, making it unlikely for the analyte to be separated by a sieving effect. The binding of MAB1 to anion exchange chromatography may be due to the zwitterionic nature of the protein. At pH 7.2, for instance, the net charge of MAB1 is positive due to its pI range of 7.5–8.6. However, if a local patch has a pI less than 6.0, it remains negatively charged at pH 7.2. This local charge patch could be the binding site to anion exchange resin. To test this hypothesis, three approaches were employed: differential chemical labeling, H/D exchange, and molecular modeling.

### 3.3. Anion Exchange Binding Site Identification by Chemical Labeling, Peptide Mapping Coupled with Mass Spectrometry

At pH 7.2, the amino acids aspartic acid (D) and glutamic acid (E) are negatively charged, with pKa values of 3.90 and 4.07 of carboxylic side chains, respectively [41]. If the carboxylic acids in MAB1 were differentially labeled at the anion exchange binding sites, this labeling could hinder their binding ability to anion exchange resins but not to cation exchangers. Therefore, carbodiimide (EDC), a water-soluble labeling agent, was chosen to modify carboxylates in MAB1. The carbodiimide-mediated modification of carboxyl functional groups requires multiple steps (Figure 1). First, carbodiimide reacts with deprotonated carboxyl groups to yield an activated intermediate, *O*-acyl isourea, which then reacts with NHS to form a semi-stable ester. Finally, glycine ethyl ester (GEE) was added to replace NHS, forming a stable amide bond through the nucleophilic attack of the free amino group, resulting in a molecular weight increase of 85 Da. The addition of NHS in the presence of EDC (at an EDC to NHS ratio of 1:2.5) prevents the hydrolysis of *O*-acyl isourea intermediate, thereby increasing the yield. To achieve various degrees of labeling, molar ratios of EDC to protein carboxylic amino acids of 1:62, 1:1, and 10:1 were used. The reaction was conducted in phosphate buffer, pH 7.0, at room temperature to ensure protein stability in more favorable conditions. Initially, the chemical labeling was intended to be performed while the protein was bound to the AEX resin as described by Dismer and Hubbuch [26]. However, the protein was eluted from the column when the chemical reagents were applied for labeling, making Dismer and Hubbuch’s approach unsuitable for this study.

The labeled MAB1 was analyzed by peptide mapping coupled with tandem mass spectrometry analysis (LC-MS/MS). The modified peptides were prominently visible in the chromatogram at a 10:1 reagent to carboxylates ratio sample (Figure 9). Among the three ratio conditions, only the 1:62 ratio sample was retained on the ProPac WAX-10 column, indicating that both the 1:1 and 10:1 EDC-to-antibody carboxylate ratios disrupted the interactions between MAB1 and AEX resin. Conversely, all three samples were retained in the WCX column, but the surface charge of the 10:1 ratio sample appeared to be disturbed, resulting in indistinguishable peaks. Additionally, only the 1:62 ratio sample retained binding activity (~60%) in the Biacore binding assay, while the other two samples showed basal activities (~5%), suggesting that the chemical modification on MAB1 may be around the CDR areas.

Three peptides (Table 2) were identified unequivocally by mass spectrometry with accurate masses and MS/MS spectra with excellent *b* and *y* ions coverage (Figure 10). All native and modified (+85 Daltons) peptides were identified by SEQUEST software and confirmed with manual inspection. For peptide 2, either E102 or D104 is possibly labeled due to the close proximity of E and D and the MS/MS spectra data that cannot clearly differentiate these two possibilities. These three peptides are found in CDR2 and CDR3 in the variable region of MAB1 with high solvent accessibility. Among these peptides, peptides 2 and 3 were modified about 50–70%, whereas peptide 1 was approximately 15%. The data suggested that these three carboxylate residues are likely involved in anion exchange binding.

Although carbodiimide labeling is carboxylate-specific, the products may not be homogeneous. The primary amines in protein itself may react with the semi-stable NHS ester before reacting with the glycine ethyl ester. This may explain, in part, many additional peaks in peptide mapping analysis compared to the control, including the light chain N-terminal peptide.

### 3.4. Anion Exchange Binding Site Identification by H/D Exchange Experiment

H/D exchange mass spectrometry was used to map the binding site(s) of MAB1 on anion exchanger Q Sepharose Fast Flow resin. When deuterium-exchanged MAB1 is loaded onto Q Sepharose Fast Flow resin, the regions that are involved in the binding are protected from exchange with bulk hydrogen molecules in the solvent. Such regions can be located by measuring deuterium levels in peptic peptides of MAB1 after the *in situ* H to D exchange followed by D to H exchange on Q Sepharose Fast Flow resin (Figure 2). A control sample was prepared by H/D exchanging the protein on Q Sepharose Fast Flow resin and then washing deuterium off from the protein sample. If the wash is complete, no deuterium should remain in any region of the protein in this control sample. By comparing deuterium levels between the deuterium-exchanged sample and the control, the binding interface between MAB1 and Q Sepharose Fast Flow resin can be identified, assuming that the binding is purely electrostatic and there are no binding-induced conformational changes.

The differences in deuteration level of MAB1 between the exchanged and control samples were plotted at four time points, 150, 500, 1500, and 5000 s. Among all quality peptides, only two light chain peptides LIYDASSL (47–54) and ESGVPSRFSGSGSGTDFTLT (55–74) showed significant deuterium incorporation (Figure 11). The remaining deuteration level was more pronounced at longer exchange times. These results suggest that the interaction with Q Sepharose FF resin involved light chain peptides (47–54 and 55–74). This is in line with the results from chemical labeling, which indicated that light chain peptide 46–61 plays a role in binding. It is worth noting that the peptides analyzed in this study provided approximately 60% sequence coverage for the heavy chain and 65% for the light chain peptides. Despite the suboptimal coverage, the H/D exchange data validated the chemical labeling results for light chain region 46–61, where D50 was labeled. The light chain peptide 55–74 may not be directly involved in the interaction with resin as chemical labeling showed less than 1% modifications on E55. It is possible that the light chain peptide 55–74 was protected from off-exchange (D to H) due to its close proximity to D50. No deuteration change was detected for the heavy chain regions 39–58 and 99–124, despite being labeled in the chemical modification experiment. This discrepancy could be due to incomplete digestion by pepsin under the test conditions and/or insufficient sensitivity of the mass spectrometer to detect the low abundant peptic peptides. Using different orthogonal techniques could provide complementary structural insights into the binding interface.

### 3.5. Predicted Structural Model Explains Residues Involved in Anion Exchange Column Binding

To confirm the involvement of the acidic residues identified through both chemical labeling and H/D exchange experiments, a structure model for the Fv region, which includes VL and VH domains of MAB1, was generated by homology modeling with the PDB coordinates 2R56 and 3D69, respectively. The VL and VH sequences of the MAB1 antibody share between 83–90% sequence identity with the selected structural templates. The surface of the total Fv area is decorated with two unique negative-charged patches (Figure 12). One of the negative surfaces is situated at the antigen binding side at the ‘top’ portion of the Fv. This more extensive negative area encompasses four out of six CDRs, namely LCDR2 and HCDR1, 2, and 3, from both VL and VH domains (Figure 12). Two acidic residues, D50 from LCDR2 and E102 from HCDR3, at the interface between VL and VH, are located close to the center of this large negative-charge antigen binding site with predicted solvent-accessible surface areas of 64 Å^2^ and 121.5 Å^2^, respectively. The strong acidic nature and the large surface areas lead to easy access to these residues by positive-charged resins. This explains how both D50 (L) and E102 (H) are involved in the interaction with anion exchange resin from chemical modification and H/D exchange experiments. A less prominent negative pocket is located at the ‘bottom’ of the Fv structure, facing towards the constant regions of the Fab structure (Figure 12). The loop stretches in the heavy chain framework 2 (HFR2) contribute predominantly to this area with two acidic residues, E43 (H) and E46 (H). E43 (H) is at the turning point of the loop nestled between two flexible residues G42 (H) and G44 (H). It rises as a ‘peak’ from an otherwise shallow pocket with a predicted large solvent-accessible surface area of 120.1 Å^2^. However, E46 (H) is located at the start of a β strand following the C-terminal end of the loop with only moderate exposure to the solvent (67.8 Å^2^). Moreover, this residue may form ion-pair interaction with R38 (H). The differences in the location, the level of solvent accessibility, and local environment between these residues from the predicted model is consistent with our observation that E43 (H), but not E46 (H), is involved in the interaction with anion exchange resin. In summary, our analysis of the predicted MAB1 model provides a structural basis on the involvement of these residues in anion exchange column interaction.

## 4. Conclusions

MAB1, a human IgG_1_ monoclonal antibody produced in CHO, shows charge heterogeneity due to N-terminal pyroglutamate formation, succinimide intermediate, deamidation, and C-terminal lysine. There is no sialic acid present in MAB1 to contribute to its charge heterogeneity. MAB1 has a unique property that the protein binds to both a cation exchanger and anion exchanger at the same neutral pH (7.2 ± 0.1), although the net charge is still positive according to its apparent pI values (7.5–8.6). Three approaches, i.e., differential ECD labeling, H/D exchange, and protein modeling, were implemented to address this unique characteristic. The results presented in this study demonstrate that several peptides located in the complementarity-determining region (CDR) or the framework of MAB1 may be involved in its interactions with anion exchange resin. Among these peptides, several carboxylate residues, e.g., E43, D50, and E102, are potentially involved in its interaction with anion exchange resins. These experimental results were strongly supported by molecular modeling. It is not a common feature for mAbs that the solvent-exposed carboxylate residuals may cause unexpected interactions with anion exchange resin, leading to further adjustment in the downstream purification scheme. The methodologies shown here may be able to extend to the study of protein binding orientation to resins in column chromatography.

## Figures and Tables

**Figure 1 antibodies-13-00052-f001:**
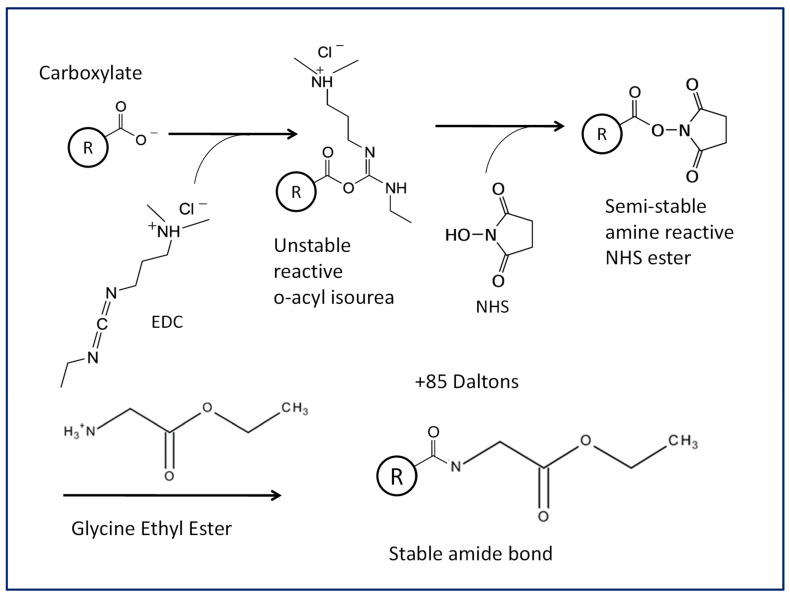
The carboxylate chemical labeling scheme for monoclonal antibody MAB1.

**Figure 2 antibodies-13-00052-f002:**
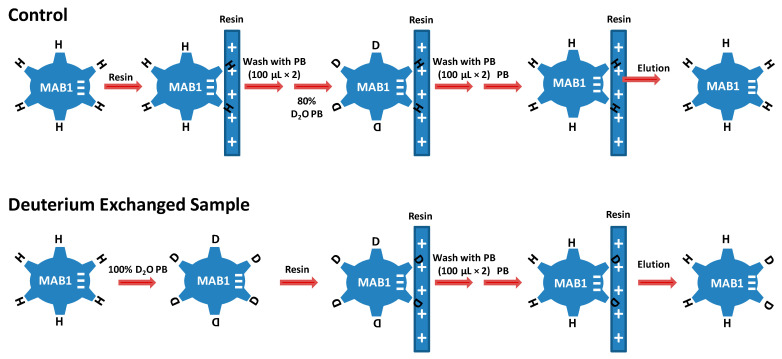
The scheme of H/D exchange experimental design. PB, phosphate buffer.

**Figure 3 antibodies-13-00052-f003:**
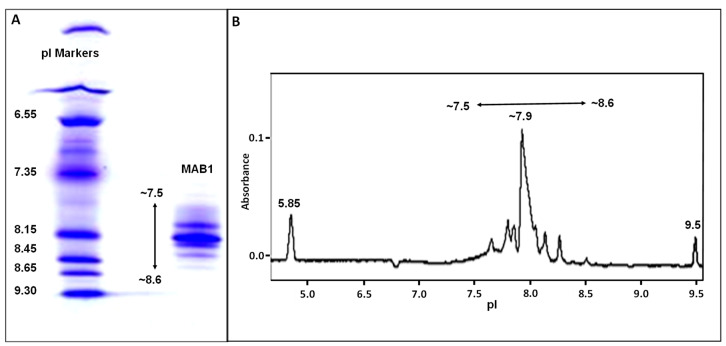
Charge heterogeneity of MAB1 by IEF gel electrophoresis (**A**) and by imaged capillary IEF (**B**). The pI makers are shown for the IEF and cIEF.

**Figure 4 antibodies-13-00052-f004:**
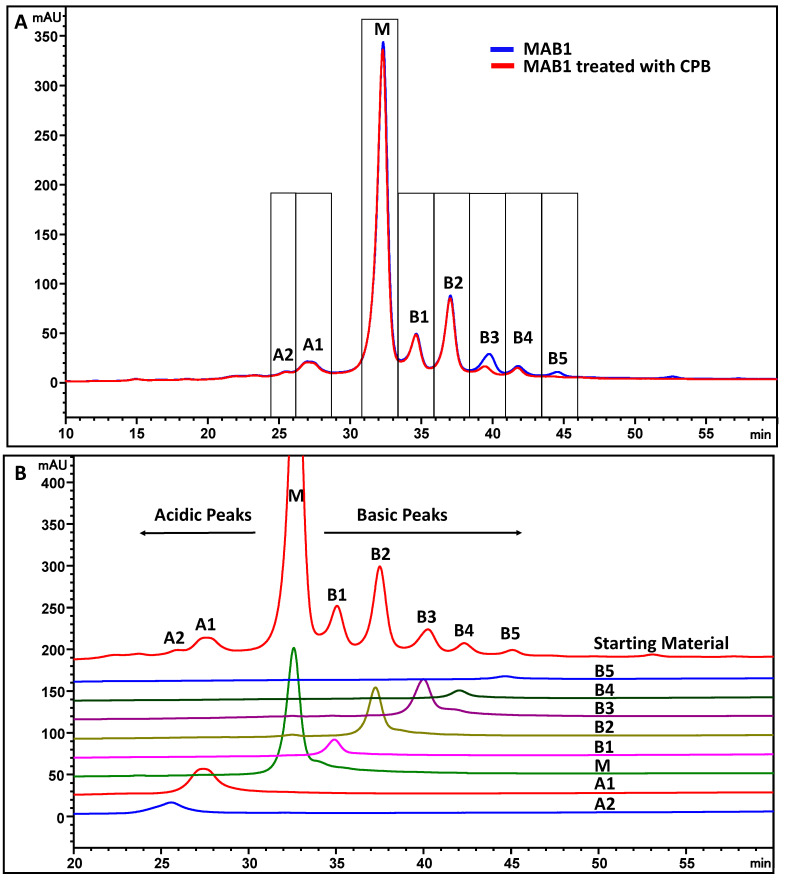
MAB1 charge variants were separated by cation exchange chromatography (**A**) and the purity of each fraction (**B**).

**Figure 5 antibodies-13-00052-f005:**
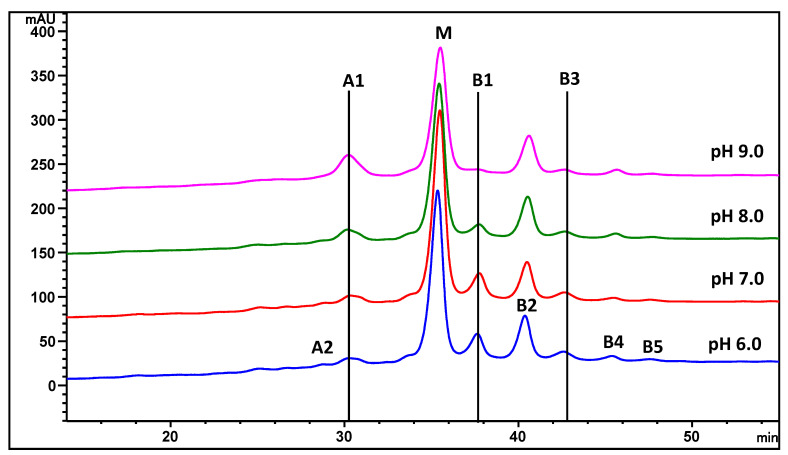
The CEX chromatograms of MAB1 after pH treatment at 37 °C overnight. Fraction numbers are the same as in Figure 4.

**Figure 6 antibodies-13-00052-f006:**
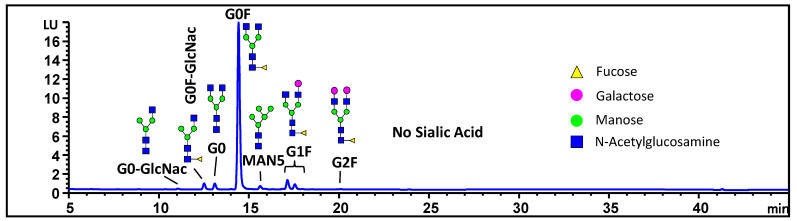
The N-linked oligosaccharide profiling of MAB1 using hydrophilic interaction chromatography.

**Figure 7 antibodies-13-00052-f007:**
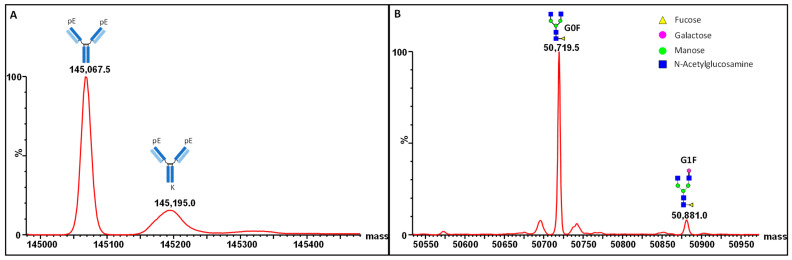
Mass analysis of MAB1 using SYNAPT G2-Si mass spectrometry (Waters). (**A**) The intact masses of deglycosylated MAB1 without C-terminal lysine and with one C-terminal lysine. (**B**) The heavy chain masses of MAB1 with G0F and G1F glycoforms.

**Figure 8 antibodies-13-00052-f008:**
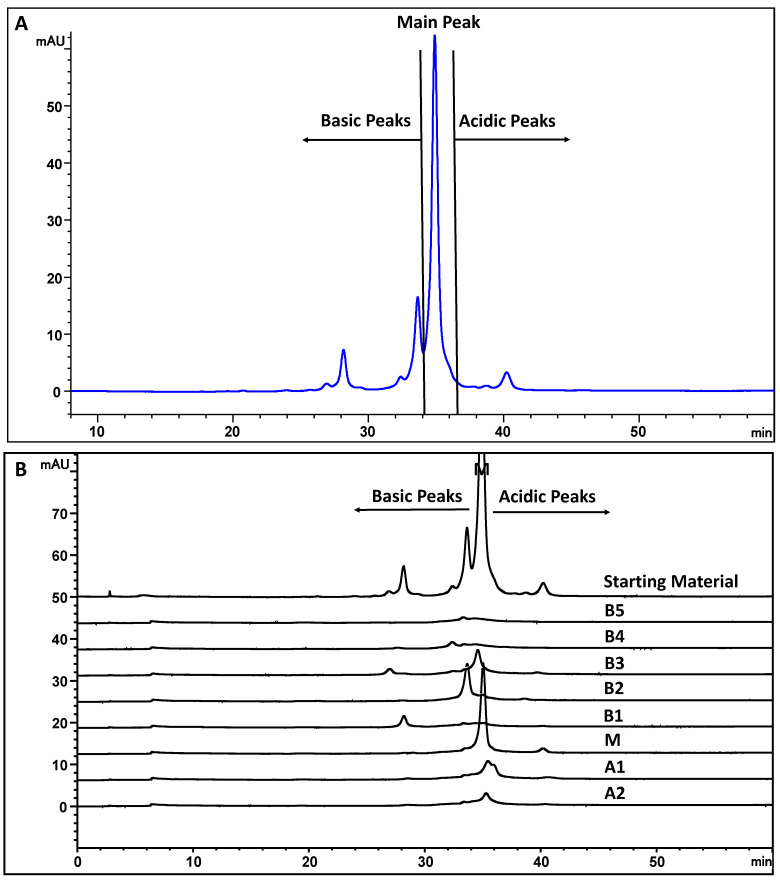
The AEX chromatograms (**A**) and AEX chromatograms of MAB1 CEX-isolated fractions (**B**).

**Figure 9 antibodies-13-00052-f009:**
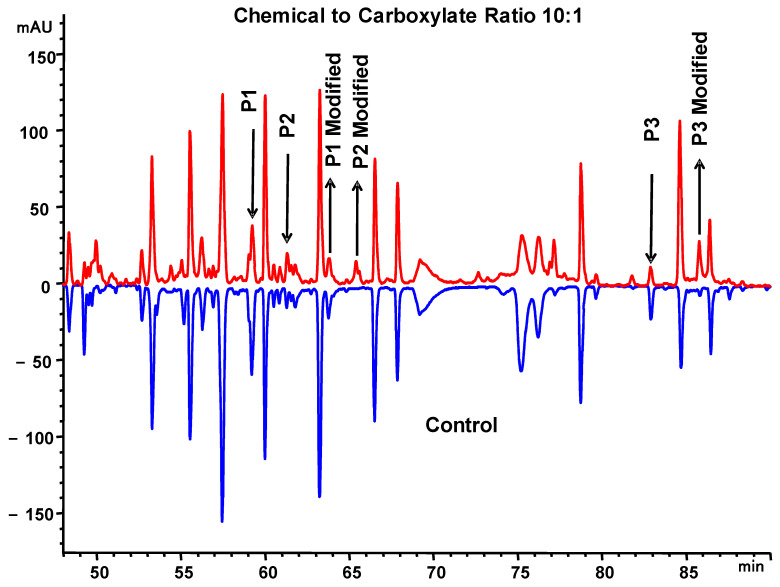
Mirror images of peptide mapping of monoclonal antibody MAB1. (**A**) Control; (**B**) chemical modification sample with chemical to carboxylate ratio of 10:1.

**Figure 10 antibodies-13-00052-f010:**
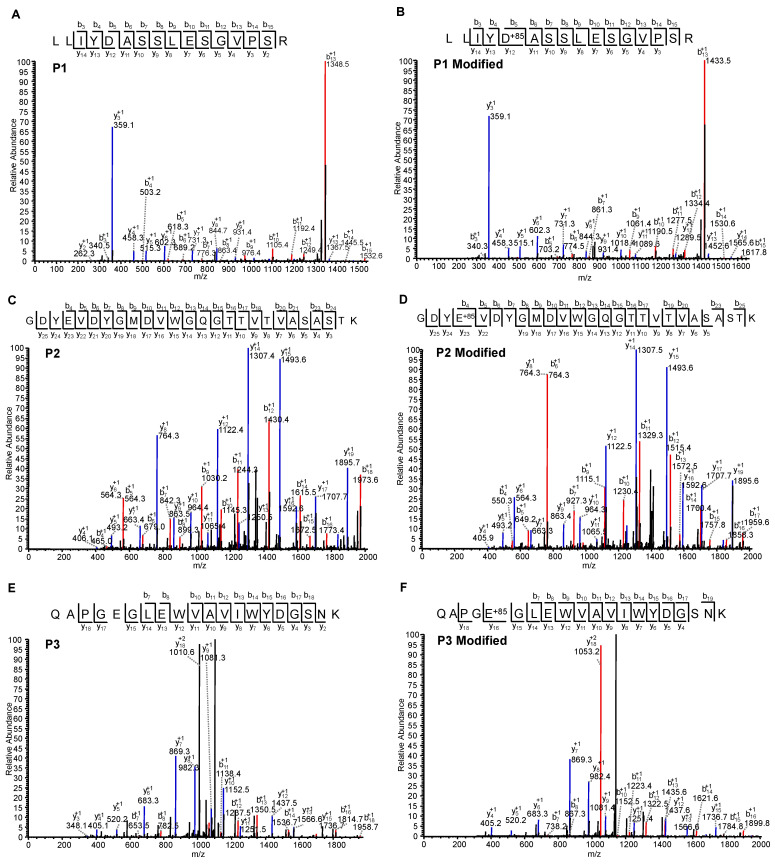
Collision-induced dissociation (CID) MS/MS spectra of MAB1 peptides. (**A**) Native peptide 1; (**B**) chemically modified peptide 1; (**C**) native peptide 2; (**D**) chemically modified peptide 2; (**E**) native peptide 3; (**F**) chemically modified peptide 3. Blue trace represents y ions and red trace represents b ions.

**Figure 11 antibodies-13-00052-f011:**
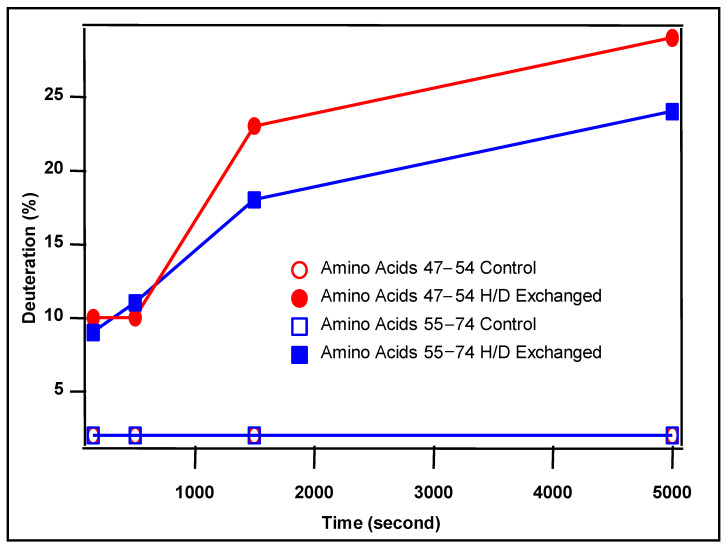
Deuteration percentage of two light chain peptides, 47–54 and 55–74. Peptide sequences are in the text.

**Figure 12 antibodies-13-00052-f012:**
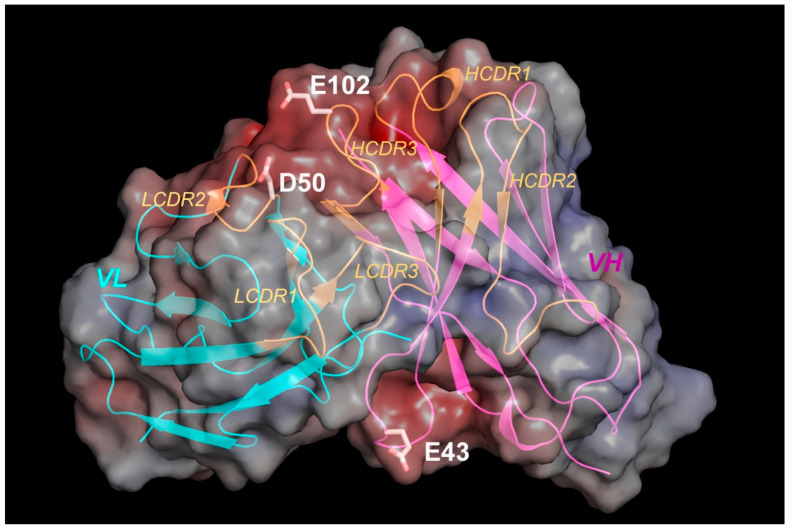
Structural model of MAB1 Fv in electrostatic surface and cartoon representations. The molecular surface is colored by the electrostatic potential (red, negative; blue, positive), starting at 10 kT/e. Light chain variable domain (VL): cyan; heavy chain variable domain (VH): magenta; CDRs (based on Kabat database): orange. Negative-charged residues with confirmed chemicals.

**Table 1 antibodies-13-00052-t001:** The post-translational modifications of MAB1 that may alter charge heterogeneity analyzed by peptide mapping coupled with mass spectrometry.

Sequence	Modification	Modified AAs	Position	CEXA2	CEXA1	CEXM	CEXB1	CEXB2	CEXB3	CEXB4	CEXB5	CRTL1	CRTL2
				(%)	(%)	(%)	(%)	(%)	(%)	(%)	(%)	(%)	(%)
QDQLVESGGGVVQPGR				1.6	1.5	0.4	1.5	36.9	13.7	51.4	33.2	8.8	8.9
	pyro-Glu	Q	H1	98.4	98.5	99.6	98.5	63.1	86.1	48.5	66.7	91.1	91.1
SLSLSPG	*Des*-K			99.0	99.4	100	98.6	99.1	62.3	86.4	60.7	95.3	95.4
SLSLSPGK				0.9	0.5	0	1.3	0.8	36.8	13.1	38.4	4.5	4.4
SLSLSP	Amidation	P	H448	0.1	0.1	0	0.1	0.1	0.9	0.5	0.9	0.2	0.2
GDYEVDYGMDVWGQGTTVTV ASASTK	Succinimide	D	H100	1.0	0.4	0.1	10.7	0.8	2.6	2.7	2.2	1.4	1.3
GFYPSDIAVEWESNGQPENNYK	Succinimide	N	H387	1.4	1.2	1.2	1.4	1.4	1.5	1.4	1.6	1.6	1.5
	Deamidated	N	H387	1.8	7.6	0.7	1.0	0.9	0.8	0.8	0.8	1.4	1.3
	Deamidated	N	H392	2.3	6.6	0.4	0.4	0.6	0.5	0.3	0.4	0.9	0.7

**Table 2 antibodies-13-00052-t002:** Chemically modified peptides of monoclonal antibody MAB1.

Peak	Fraction	Mass_Theo_	Mass_obs_	^1^ Modified Mass_obs_	Position	Sequence
P1	LC_D50	1705.90	1705.84	1790.92	46–61	LLIYDASSLESGVPSR
P2 ^2^	HC_E102/D104	2736.22	2736.00	2821.12	99–124	GDYEVDYGMDVWGQGTTVTV ASASTK
P3	HC_E43	2218.07	2217.84	2302.92	39–58	QAPGEGLEWVAVIWYDGSNK

^1^ Chemically modified amino acid (+85 Daltons). ^2^ It cannot rule out the possibility that D104 is modified based on the MS/MS data.

## Data Availability

All data supporting the findings of this study are available from the corresponding authors upon request.

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
