# Peer review of "Characterization of the Charge Heterogeneity of a Monoclonal Antibody That Binds to Both Cation Exchange and Anion Exchange Columns under the Same Binding Conditions"

_2073-4468, 2024, doi:10.3390/antib13030052_

Round 1

Reviewer 1 Report

Comments and Suggestions for Authors

The manuscript titled “Characterization of the Charge Heterogeneity of a Monoclonal Antibody that Binds to Both Cation Exchange and Anion Exchange Columns at the Same Binding Conditions” was written well and is a good work but there are certain issues.

1.      The abstract is lengthy and unclear, making it hard to grasp the main findings. The authors are advised to shorten it and provide a clear focus for easier reader comprehension.

2.      Introduction: Authors can add a portion on how mAbs are used in therapies and their advantages over others.

3.      Lack of clarity in the preparation of control samples, including steps to remove D2O buffer, washing by phosphate buffer with 0% D2O, and quenching in the exchange reaction.

4.      Incomplete details about the maximum deuterated sample preparation, including the specific incubation conditions and the subsequent steps.

5.       The description of the preparation of labeled MAB1 and the subsequent peptide mapping coupled with tandem mass spectrometry analysis (LC-MS/MS) does not have sufficient information to interpret.

6.      Discussion: Details about the identification of the anion exchange binding site using the H/D exchange experiment are incomplete.

7.      The language can be improved. 

Comments on the Quality of English Language

A little improvement is required

Author Response

Comment 1: The abstract is lengthy and unclear, making it hard to grasp the main findings. The authors are advised to shorten it and provide a clear focus for easier reader comprehension.

Response: Thanks a lot to point it out. It was too lengthy and out of focus. The abstract has been revised for brevity and clarity, making it simpler for readers to understand the experimental methodologies and outcomes. 

Comment 2:  Introduction: Authors can add a portion on how mAbs are used in therapies and their advantages over others.

Response: Thanks a lot for the suggestion! The information was added in lines 40-42.

Comment 3: Lack of clarity in the preparation of control samples, including steps to remove D2O buffer, washing by phosphate buffer with 0% D2O, and quenching in the exchange reaction.

Response: Thanks for the comment! We revised and clarified the procedures in Section 2.10.

Comment 4: Incomplete details about the maximum deuterated sample preparation, including the specific incubation conditions and the subsequent steps.

Response: Thanks for the comment! Lines 279-283 were revised to include incubation and quenching conditions.

Comment 5: The description of the preparation of labeled MAB1 and the subsequent peptide mapping coupled with tandem mass spectrometry analysis (LC-MS/MS) does not have sufficient information to interpret.

Response: We appreciate this comment! Section 3.3 was revised. Mass spec data illustrated in Figure 10 demonstrated the EDC labeling with +85 Daltons at D or E residues.

Comment 6: Discussion: Details about the identification of the anion exchange binding site using the H/D exchange experiment are incomplete.

Response: Thanks a lot for the comment! Section 3.4 was revised to ensure 2 light chain peptides (47-54 and 55-74) are involved in AEX binding.

Comment 7. The language can be improved. 

Response: Appreciate the suggestion! The entire manuscript has been revised to improve English.

Reviewer 2 Report

Comments and Suggestions for Authors

The work provides original and interesting data that integrate physical properties of monoclonal antibodies and their practical applications. At whole, the manuscript gives clear and logical presentation of the obtained results. However, some issues need justification/revision:

1. The Abstract could be shortened by more compact description of common statements and minor features of the presented experimental data.

2. Please check the data at lines 48-58. Two quantities of approved preparations – more than 100 and nearly 120 - confuse the reader. Besides, the lists of concrete «not-including» preparations at lines 53-57 are not useful for further understanding the authors' work.

3. The long paragraph at lines 116-140 is finalized by the comment «Here we demonstrated the feasibility to use these techniques to understand the binding interactions». The addressing to «these techniques» is not clear. It would be better to finalize the Introduction by direct indication of the aim of the study and main steps in reaching this aim.

4. References to prototype techniques or to directly used techniques will be useful for all sub-sections about methods in the Section 2.

5. Fig. 5B should be given in better resolution. The limits for 7.5-8.6 range are not clear now. pI identification for main peaks would be more useful here.

6. Lines 371-372. The addressing to Table 1 concerning the value of C-terminal lysines needs more detailed justification and calculation.

7. The consideration of the charge heterogeneity of monoclonal antibodies as a factor influencing their immobilization on different surfaces in earlier works should be commented. The corresponding sub-section before the conclusion would be useful for this purpose.

Author Response

Comment 1. The Abstract could be shortened by more compact description of common statements and minor features of the presented experimental data.

Response: We appreciate the comment. The abstract was revised to be concise and focused.

Comment 2. Please check the data at lines 48-58. Two quantities of approved preparations – more than 100 and nearly 120 - confuse the reader. Besides, the lists of concrete «not-including» preparations at lines 53-57 are not useful for further understanding the authors' work.

Response: We agree the comment and appreciate it very much! The paragraph was updated and irrelevant information was removed.

Comment 3. The long paragraph at lines 116-140 is finalized by the comment «Here we demonstrated the feasibility to use these techniques to understand the binding interactions». The addressing to «these techniques» is not clear. It would be better to finalize the Introduction by direct indication of the aim of the study and main steps in reaching this aim.

Response: Thanks a lot for this comment. It was too lengthy and out of focus. We have revised the paragraph to clarify the main findings of this study.

Comment 4. References to prototype techniques or to directly used techniques will be useful for all sub-sections about methods in the Section 2.

Response: Thank you very much for the suggestion! We have some general references in the beginning of each result/discussion section. The methods used in this study were specific for each condition; we referenced some techniques.

Comment 5. Fig. 5B should be given in better resolution. The limits for 7.5-8.6 range are not clear now. pI identification for main peaks would be more useful here.

Response: Thanks a lot for the comment. It might be a typo for Figure 5B. The Figure 3B (Imaged Capillary IEF) was updated with better resolution and the main peak was labeled.

Comment 6. Lines 371-372. The addressing to Table 1 concerning the value of C-terminal lysines needs more detailed justification and calculation.

Response: Thanks a lot for the comment! C-terminal lysine was found in Peak B3 and B5 (approximately 40% based on the peptide mapping, Table 1). The percentage was calculated based on the total C-terminal peptides (SLSLSPG+SLSLSPGK+SLSLSP) However, these peaks do not consist of purely C-terminal lysine but also some succinimide intermediates. 

Comment 7. The consideration of the charge heterogeneity of monoclonal antibodies as a factor influencing their immobilization on different surfaces in earlier works should be commented. The corresponding sub-section before the conclusion would be useful for this purpose.

Response: We appreciate this comment. We added a comment in lines 678-680 to emphasize this unique feature of this MAB1 antibody.
